# Significant Response of Methane in the Upper Troposphere to Subseasonal Variability of the Asian Monsoon Anticyclone

Sihong Zhu[1], Mengchu Tao[1,2*], Zhaonan Cai[1,3], Yi Liu[1,3,4], Liang Feng[5,6], Pubu Sangmu[7], Zhongshui Yu[8], Junji Cao[9]

[1]Carbon Neutrality Research Center, Institute of Atmospheric Physics, Chinese Academy of Sciences, Beijing 100029, China
[2]Key Laboratory for Middle Atmosphere and Global Environment Observation, Institute of Atmospheric Physics, Chinese Academy of Sciences, Beijing 100029, China
[3]Key Laboratory of Atmospheric Environment and Extreme Meteorology, Beijing 100029, China
[4]University of Chinese Academy of Sciences, Beijing 100049, China
[5]National Centre for Earth Observation, University of Edinburgh, Edinburgh, EH9 3FF, UK
[6]School of GeoSciences, University of Edinburgh, Edinburgh EH9 3FF, UK
[7]Linzhi City Meteorological Bureau/CMA Mêdog Field Observatory for Atmospheric Water Cycle, Linzhi, 860000, China
[8]Xizang Institute of Plateau Atmospheric and Environmental Science Research/Xizagê National Climate Observatory, Lhasa, 850001, China
[9]Institute of Atmospheric Physics, Chinese Academy of Sciences, Beijing 100029, China

*Correspondence to*: Mengchu Tao (mengchutao@mail.iap.ac.cn)

**Abstract.** Substantial methane ($CH_4$) emissions in Asia are efficiently transported to the upper troposphere through the monsoon dynamical system, which forms a remarkable seasonal $CH_4$ enhancement in the upper troposphere. Using a chemical transport model GEOS-Chem driven by surface optimized $CH_4$ flux, the $CH_4$ enhancement over the Asian monsoon region is explored as a combined effect of the monsoon dynamical system and regionally increased emissions during late monsoon season. The spatial distributions of $CH_4$ at the upper troposphere show strong subseasonal variability, which is closely tied to the east-west oscillation of Asian monsoon anticyclone (AMA). Besides, the AMA patterns influence the efficiency of methane-rich air transport to the upper troposphere and lower stratosphere. The AMA center around 80°E favors the upward transport from organized monsoon convective sources over the Indian subcontinent and Bay of Bengal while the AMA center around 105°E favors the source from southwest China transported to the upper troposphere. When the AMA shifts over the Iranian Plateau, vertical transport from the convective outflow level further to the upper troposphere is weakened and the horizontal redistribution becomes dominant. According to our model sensitivity study, the differences in the upper tropospheric $CH_4$ anomalies caused by large-scale circulation is 1-2 times of that caused by regional surface emissions. Our research highlights the complex interaction between monsoon dynamics and surface emissions to determine the upper tropospheric methane.

## 1 Introduction

Methane ($CH_4$), the second most important greenhouse gas, emitted heavily from south Asia and China, accounts for ~25% of the global anthropogenic emission budget in the recent decades (Stavert et al., 2022). The Asian summer monsoon (ASM)

has been proven to be an efficient pathway connecting the rich methane boundary layer and the upper troposphere and lower stratosphere (UTLS) (Randel et al., 2010). The $CH_4$ enhancement in the UTLS impacts the climate through radiative forcing (Riese et al., 2012) and influences the stratospheric chemistry, e.g., the methane oxidation (Rohs et al., 2006). More critically, limited understanding of non-local methane sources might mislead the flux inversion from total columns from satellite products (Zeng et al., 2021).

Observations (satellite retrievals and in-situ measurements) and models have evidenced remarkable enhancements of tropospheric tracers over the ASM region in the UTLS, including CO, water vapor, HCN and hydrocarbons, and aerosols (Bucci et al., 2020; Park et al., 2009; Pan et al., 2016; Pan et al., 2024; Randel et al., 2010; Rosenlof et al., 1997; Yu et al., 2017). Similar like other tropospheric tracers, substantial observational evidence showed distinct spatiotemporal distribution of high methane in the middle to upper troposphere over the Asian region during the late monsoon season (Baker et al., 2012; Schuck et al., 2010; Tao et al., 2024; Tomsche et al., 2019) as well as its lasting and traceable pathways into the southern hemisphere and the stratosphere after monsoon withdraw (Belikov et al., 2022; Yan et al., 2019).

The ASM transport structure connecting the surface source regions and the Asian monsoon anticyclone (AMA) in the UTLS has been characteristically described by various studies. The transport system is marked by deep convection, rapidly injecting surface air masses up to the convective outflow level, potential temperature heights 360 K (~16 km), which commonly refers to "fast convective chimney" (Pan et al., 2016). Above 370 K, the continuous upward transport is mainly an anticyclonic "spiraling" movement (Bergman et al., 2013; Lau et al., 2018; Ploeger et al., 2017; Vogel et al., 2019) along with a slow upwelling in a vertical velocity typically 0.3~0.8 K/day (Garny and Randel, 2016; von Hobe et al., 2020; Legras and Bucci, 2020; Vogel et al., 2024). Significant deep vertical transport is predominantly observed in the southeastern quadrant of the anticyclone, centered near the southern flank of the Tibetan Plateau. This "chimney-like transport" actively uplifts air from the highly-polluted regions, like northeast India, Nepal, and the northern Bay of Bengal, to the upper troposphere. This rapid convective uprising process was further characterized by a 'double-stem-chimney cloud' (Lau et al., 2018) or 'two-stem mushroom' (Pan et al., 2022) structure, which indicates two key areas prominently contributing to the "fast convective chimney": the Himalayas-Gangetic Plain and the Sichuan Basin in southwestern China.

One key dynamical feature for the ASM is its subseasonal variability, which is characterized by the active–break cycle of precipitation and surface pressure patterns (Krishnamurti and Ardanuy, 1980; Krishnamurti and Bhalme, 1976), as well as the east-west oscillation of AMA (Nützel et al., 2016; Zhang et al., 2002) with eddy shedding of low potential vorticity (PV) air in the UT (Garny and Randel, 2013). The variability in the monsoon convection pattern potentially modulates the intensity and position of the AMA (Garny and Randel, 2013; Nützel et al., 2016; Siu and Bowman, 2020). Moreover, the combined effects of convective uplift and anticyclonic confinement jointly shape the distribution and transport of tracers such as CO in the UTLS, as demonstrated and analysed in previous studies (Luo et al., 2018; Pan et al., 2016).

According to simulations with chemical transport models as well as the satellite observations, it is evident that the $CH_4$ in the upper troposphere (UT), similar as tracer like CO and water vapor, also has an isolated center over Asian monsoon region

during the boreal summer (Park et al., 2004, Tao et al., 2024; Xiong et al., 2009). This increase of $CH_4$ is about 100 ppb higher than its regional annual mean volume mixing ratio (VMR) and is 3%~10% higher than the VMRs averaged over the same latitude (Tao et al., 2024; Xiong et al., 2009). The relevant studies rarely use $CH_4$ as monsoon transport tracer due to its complicated emission sources. Different from tracer like CO, that has primarily industrial sources and hence displays little seasonal variation, $CH_4$ emissions in Asia, predominantly (over 60%) from rice cultivation (Stavert et al., 2022), exhibit significant seasonality. The debate persists over which factor plays the dominant role in the seasonal increase of UT methane over the Asian monsoon region—enhanced summer emissions from regional rice paddies (Zhang et al., 2020) or the strong upward transport by the monsoon convection and circulation (Zeng et al., 2021) —as both are known to contribute. Hence, it is crucial to understand how the lower boundary methane conditions and monsoon circulation interact with each other.

The purpose of this study is (1) to explore the association of UT methane over ASM with subseasonal variability of AMA dynamics and (2) quantify the relative role of the AMA dynamics and regional emissions in shaping UT methane. The state-of-the-art approach for the goal is model simulations with a data assimilation system. The reasonable representation of UT $CH_4$ over ASM with this model has been proven through a comprehensive comparison with satellite and in-situ observations in a previous study (Tao et al., 2024). In this study, we first show the subseasonal behaviour of UT methane modulated by the AMA using the case of summer 2020. Then, we analyze the methane transport pathways under different AMA modes through mode composites. Lastly, we examine the UT methane change in association with AMA dynamics and surface emission respectively, through a model sensitivity study.

## 2 Data and Method

### 2.1 Global 3-D methane simulation

We use v12.5.0 of GEOS-Chem (http://www.geos-chem.org, The International GEOS-Chem User Community, 2019) to generate of global 3-D methane concentrations for the study period 2015–2020, with a temporal resolution of 1 day and a spatial resolution of $2° \times 2.5°$ (longitude × latitude). The model is driven by the Modern-Era Retrospective Analysis for Research and Applications version 2 (MERRA-2) re-analysis from the Global Modelling and Assimilation Office of NASA (Gelaro et al., 2017), which also provides the convective mass fluxes used to represent convection. For consistency, we also use MERRA-2 reanalysis data to analyze dynamical fields.

The surface $CH_4$ fluxes used in the transport model are optimized with atmospheric observations via an ensemble Kalman Filter (EnKF) framework (Feng et al., 2009, 2017). In the framework, the *a priori* emission estimates include natural sources (e.g., wetlands, fires, termites), and anthropogenic sources (e.g., fossil fuels, livestock, rice, and waste) as detailed in Zhu et al., (2022). The atmospheric $CH_4$ observations include the proxy GOSAT v9.0 column methane data ($XCH_4$) from the University of Leicester (https://catalogue.ceda.ac.uk/uuid/18ef8247f52a4cb6a14013f8235cc1eb/, Parker et al., 2020) and near-surface methane mole fraction samples from the $CH_4$ GLOBALVIEWplus v5.0 ObsPack

The inverted flux-driven global 3-D methane concentrations were evaluated using observations from several platforms, including ground-based $XCH_4$ measurements from the Total Carbon Column Observing Network (TCCON) (https://tccondata.org/, TCCON Team, 2022), $CH_4$ flask samples collected by aircraft in the Comprehensive Observation Network for TRace gases by AIrLiner (CONTRAIL) project (https://www.cger.nies.go.jp/contrail/index.html, Machida et al., 2023) and $CH_4$ profiles from ground to 25 km measured over the Tibetan Plateau by an AirCore air sampling and data processing system (Tao et al., 2024). In our simulation experiments, only surface $CH_4$ flux is varied, with no alterations to meteorological fields or initial conditions.

## 2.2 Classification of Asia monsoon anticyclone modes

In a series of previous studies, the subseasonal variability of AMA has been mainly characterized as 'bimodality', referring to two major modes: Tibetan mode and Iranian mode (Nützel et al., 2016; Zhang et al., 2002). It is important to note that the existence and nature of this bimodality remain a topic of ongoing debate. For instance, Nützel et al. (2016) demonstrated that the bimodal distribution of the AMA center is sensitive to the choice of reanalysis datasets. Manney et al. (2021) also proposed that the bimodality is not a prominent feature for the AMA climatological state. Our study does not aim to confirm the existence of AMA bimodality or to explore its underlying dynamics. We aim to investigate the relationship between AMA subseasonal variability and the three-dimensional distribution and transport of methane.

To identify the AMA major center, we use wind and geopotential height (GPH) data from MERRA-2 reanalysis and basically follow the methodology outlined in Zhang et al. (2002) as shown in Figure 1. The AMA centers are identified as the positions of GPH peaks (magenta crosses in Fig.1) along the ridgeline of anticyclone (zero u-wind within the ASM region, i.e. 30°W-180°E, 10°N-40°N, shown as blue dotted line in Fig. 1) on daily basis. We exclude centers with GPH values in the lowest 70% of the GPH range measured along the ridgeline. Different from previous studies that commonly use 100 hPa as a reference, we use the 150 hPa pressure level because both convection-driven vertical uplift and horizontal confinement are active at this layer, providing significant variability in AMA and its impact on transport of tracer gases. Meanwhile, the anticyclonic circulation basically remains closed at 150 hPa.

As shown in Fig. 1(d), our statistics on the probability distribution of AMA center positions is based on days with a single center during 6 boreal summers (July, August and September, JAS) from 2015 to 2020. The distribution exhibits a pattern resembling bimodality, though it is less distinct than that reported in Zhang et al. (2002). The frequency of the Tibetan mode (the eastern phase of the distribution) is nearly twice that of the Iranian mode. Furthermore, within the Tibetan Plateau (TP) cases, we find a weaker bimodal structure with one peak located to the west of 90°E and another to the east.

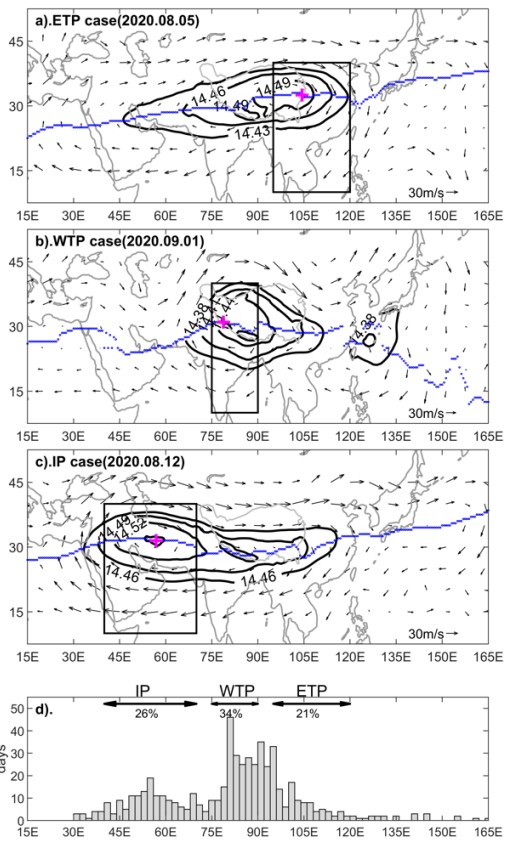

**Figure 1: The daily meteorological maps are shown as three mode cases in panels (a-c), including horizonal wind field (arrows) and geopotential height (GPH, black contours) on 150 hPa. The blue dotted line denotes the zero u-wind, indicating the ridgeline of anticyclone. The magenta cross marks the local GPH peaks along the ridgeline, which is identified as AMA center. The distribution of the AMA major center along the longitudes is shown in panel (d). This statistic includes the daily meteorological data covering 6 boreal summers (July, August and September) from 2015 to 2020. The frequencies of occurrence (%) for the three AMA modes are also labeled. The rectangle in (a-c) outlines the geographic domain corresponding to each mode.**

Nützel et al. (2016) proposed that due to the continuous spatial variability of the AMA center and its connection to convective activity, a more flexible classification approach should be adopted rather than relying on a fixed bimodal structure. Variations in regions of enhanced convection and associated changes in AMA morphology potentially influence the distribution of upper tropospheric tracers through vertical transport and anticyclonic confinement (e.g. Pan et al., 2016; Tomsche et al. 2019; Vogel et al., 2019). Based on this perspective, our study further subdivides the Tibetan Plateau (TP) mode into the East Tibetan Plateau (ETP) and West Tibetan Plateau (WTP) modes. Consequently, we define three AMA modes in total: the Iranian Plateau (IP) mode (40–70°E), WTP mode (75–90°E), and ETP mode (95–120°E).

## 3 Results

### 3.1 Subseasonal variability in upper tropospheric CH₄ during 2020 Asian summer monsoon

We firstly examine the covariability of the large-scale circulation of ASM and the horizontal distribution of CH₄ at 150 hPa using the case in the summer of 2020. Figure 2 (a-b) presents Hovmöller diagrams of GPH and CH₄ anomalies on 150 hPa referring to their respective daily means averaged over the core ASM region (15°N–40°N, 15°E–135°E). The AMA experienced four east-west oscillations from July to mid-August and the AMA center remained predominantly east of 75°E from mid-August onwards (Figure 1a). The simulated CH₄ anomalies display a strong spatial correlation with the GPH field,

with a Pearson correlation coefficient of 0.78. This subseasonal variability of AMA significantly modulated the CH₄ variations in the middle to upper troposphere (see Figure 1b), similar to the behavior of tracers such as CO (Pan et al., 2016), primarily due to anticyclonic confinement (Ploeger et al., 2015).

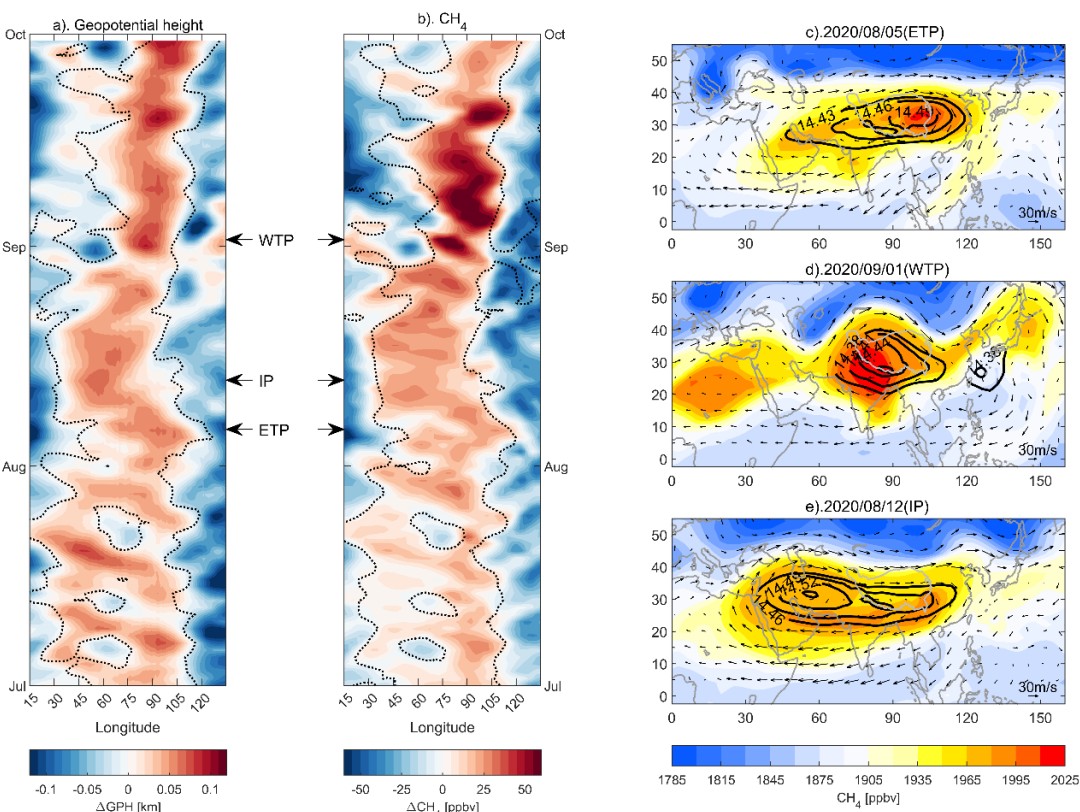

**Figure 2: Hovmöller diagrams of anomalies in (a) geopotential height and (b) CH₄ concentrations at 150hPa for JAS 2020.**
**Anomalies are calculated with respect to the daily mean values averaged over the main ASM region (15°N–40°N, 15°E–135°E). The zero-contour (dotted line) of each panel is plotted on opposite filed for reference. The Pearson correlation coefficient between the two anomaly fields is 0.78. Panels (c-e) show maps of daily mean CH₄ concentration at 150hPa (color shades) for three selected days (indicated by arrows in the Hovmöller diagrams), representing characteristic states for the WTP, IP and ETP mode of AMA subseasonal oscillation. Black contours denote selected geopotential height levels highlighting the AMA center, and vectors**
**represent the horizontal wind field at 150 hPa.**

The east-west migration of CH$_4$ anomalies mirrors the subseasonal oscillations of the AMA on daily basis, revealing the dynamic nature of methane distribution over monsoon region. Hereby we select 3 days to illustrate the spatial distribution of methane and dynamical fields under the three AMA modes in Fig. 1(c-e): East Tibetan Plateau (ETP) mode centering 90°E-110°E (Aug. 5th), West Tibetan Plateau (WTP) mode centering 80°E-90°E (Sep. 1st) and Iranian Plateau (IP) mode centering 50°E-60°E (Aug. 12th). The daily maps show that although the high CH$_4$ was largely confined within AMA, CH$_4$ peak does not invariably align with the AMA center. For example, at IP mode on Aug. 12$^{st}$, the high methane center locates southern edge of AMA. This pattern results from the "stirring" interaction between convection-uplifted boundary layer air from the Indian subcontinent (as shown by the overlap between the main monsoon convection source and high methane regions near the lower boundary in Fig. S1) and the surrounding air, which is similar like "stirring" interaction proposed by Pan et al. (2016).

The subseasonal dynamical control of AMA on CH$_4$ in the middle to upper troposphere is similar as other tracer like CO (Luo et al., 2018; Pan et al., 2016). Contrasting with CO—which typically shows a significant increase in UT around mid-June, correlating with the onset of the South Asian monsoon and its associated convective transport—we observe a pronounced increase in CH$_4$ concentrations at UT starting from mid-August, e.g. notably CH$_4$ increase within the 75°E-110°E longitudinal range in Fig. 1(b). The timing of this CH$_4$ surge aligns closely with the seasonal emission peak mainly from the rice paddy cultivation as suggested by Zhang et al. (2020).

Observed the methane behavior throughout this summer, the monsoon large-scale circulation dynamics with a strong subseasonal variability clearly manipulate the spatial distribution of upper tropospheric CH$_4$, which is similar as tracers like CO and water vapor. Meanwhile, the temporal subseasonal variation of the mean CH$_4$ concentrations over AMA (a remarkable integral rise during mid-August and September) potentially relates to increased emissions in late monsoon season.

### 3.2 Composites of AMA modes

To extend the 2020 summer case to a general picture of CH$_4$ distribution under different AMA status, we composite the methane daily fields for 6 summers (JAS during 2015-2020) according to the corresponding AMA modes (see the Method section for the classification). According to the limited statistical samples, we found that IP mode predominantly manifests during Julys, while the ETP mode is more frequently observed in Septembers. This phenomenon is also observed in the 2020 case as shown in Figure 1. Figure 3 presents the vertical structure of methane (longitude-pressure cross section) associated with three single-center modes (WTP, ETP and IP mode). It reveals that high concentrations of methane in the boundary layer are localized between 60°E to 105°E and undergo a redistribution to a broader area in the upper troposphere (300-100hPa).

This distribution emerged with the "double-stem-chimney cloud" or "two-stem-mushroom" structure, as characterized by (Lau et al., 2018; Pan et al., 2022), i.e., the two narrow "stems" (centered on 80°E and 105°E, respectively) indicating vertical pumping by deep convections as well as the "cap" resulting from the expanding the uplifted near-surface air masses

through quasi-isentropically transport. The composites add details to the "two-stem-mushroom" structure. The WTP mode favors the vertical uplifting over 80°-90°E (western "stem") while the ETP mode favors the vertical transport over 100°-110°E (eastern "stem"). Previous study has identified two key regions in connection with these two transport "stems": one is the Himalayas-Gangetic Plain mainly including the northern Indian region and Bangladesh, i.e. Indian subcontinent, and another one is southwest China (Lau et al., 2018, see also the surface flux map for monsoon season in Figure S1). As shown in Fig. 3(c), CH₄ enhancement in the upper troposphere under the IP mode is weaker than in the WTP and ETP modes. The underlying reason is not straightforward; it may result from weaker vertical uplift due to weaker convection in IP mode, a shift of the main convection to regions with lower emissions, and/or the position of the AMA center favoring horizontal redistribution over further vertical upwelling.

To further explore this, Figure 4 presents horizontal views at multiple layers to illustrate the relationship between convective sources and CH₄ enhancement in the upper troposphere for each AMA mode. Note that the CH₄ enhancement in Fig. 4 is expressed as a percentage (%) relative to the global zonal mean at each latitude band within each pressure layer. The corresponding CH₄ VMR absolute values can be found in Fig. S2.

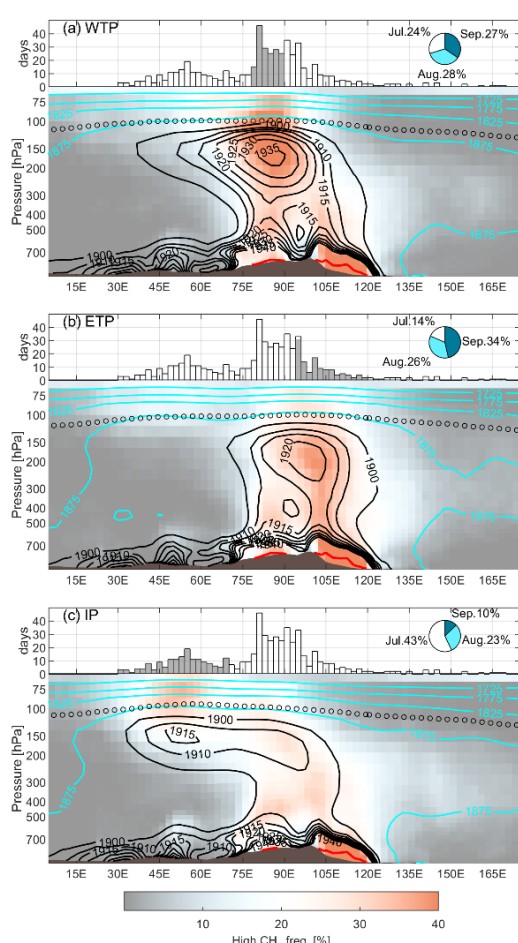

**Figure 3: Composites of the longitude-pressure cross section of methane (averaged over 15°N-40°N) for the three modes (2015-2020 JAS). The contours show the averaged methane volume mixing ratios (VMRs) for each mode (VMRs less than 1900ppbv shown as cyan contours and VMRs equal or greater than 1900ppbv shown as black contours). The color shades indicate the occurrence frequency (unit: %) of high $CH_4$ VMR (criteria: higher than 90% of grids within the Asian monsoon region on the same model level). The longitude distribution of GPH maximum along the anticyclone ridge is plotted on the top of each panel, where the grey bars cover the longitude range for corresponding mode. The pie charts show the proportions of months for each mode.**

As being suggested by Fig. 3, the horizontal maps confirm that the WTP mode facilitates the vertical transport from southern side of TP to the UT (see Fig. 4 left column, a1-a3), whereas the ETP mode is more conducive to promoting the vertical transport over eastern side of TP (Fig. 4 middle column, b1-b3). The IP mode composite demonstrates that while UT methane enhancement is observed between 45°E to 60°E (Fig. 4 right column, c3), the corresponding enhancement at the middle troposphere is located similarly to the TP modes, i.e., at the southern flank of the Tibetan Plateau (Fig. 4 c3).

Comparing the three bottom panels, we find that differences in $CH_4$ distribution within the lower troposphere among the three modes are minor. The occurrence of deep convection (shown as magenta contours, outgoing longwave radiation less than 210K) together with high methane emissions (bottom row in Fig. 4) jointly determine the methane distribution in the mid-troposphere (third row in Fig. 4). For example, in both the WTP and IP modes, deep convection is enhanced over the southern flank of the TP, where methane emissions are intense, resulting in a mid-tropospheric methane enhancement in this region (Fig. 4 a3 & c3). In contrast, during the ETP mode, convection is suppressed over the Indian subcontinent but get active over southwestern China, causing the high methane center to shift to southwestern China (Fig. 4b3). It is also notable that these convective sources with elevated methane are subject to redistribution and confinement by the anticyclonic circulation in the upper troposphere (100–300 hPa).

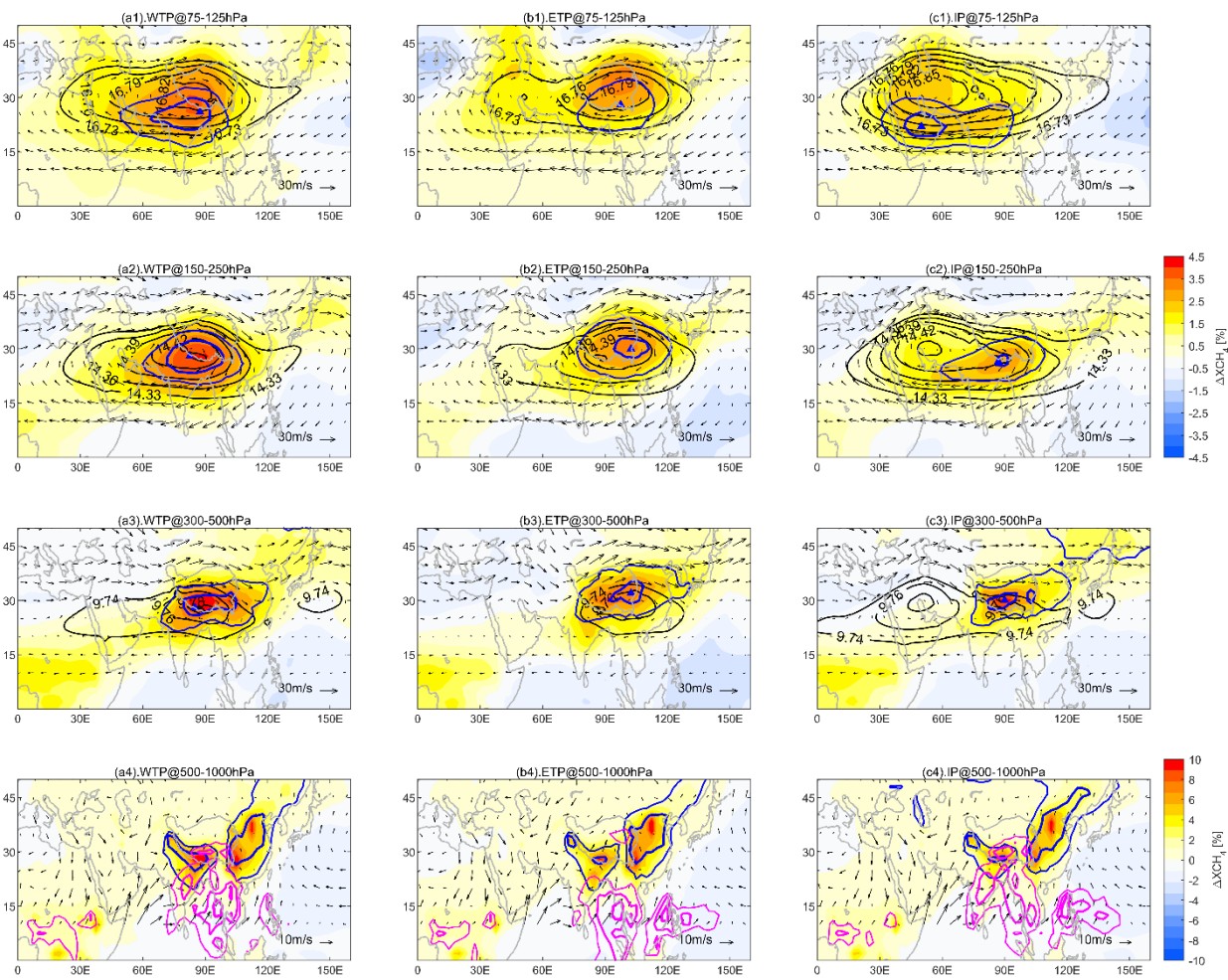

**Figure 4: Horizonal distribution of XCH₄ anomaly composites corresponding to the three dominant AMA modes, shown across 4 vertical layers: the lowermost stratosphere (75–125 hPa; top panels a1–c1), the upper troposphere (150–250 hPa; panels a2–c2), the middle troposphere (300–400 hPa; panels a3–c3), and the lower troposphere (500–1000 hPa; bottom panels a4–c4). Color shading indicates the XCH₄ anomalies, expressed as a percentage (%) relative to the global zonal mean at each latitude within each pressure layer. Black contours depict the GPH field outlining the AMA structure. Blue lines enclose regions with high occurrence frequencies (50% and 70%) of elevated methane—defined as in Figure 3—and triangles mark the locations with highest occurrence frequency. Magenta contours indicate outgoing longwave radiation (OLR) composites lower than 210 K for each mode, with an interval of 10 K, which represents the deep convective cloud.**

It is suggested that the spatial relationship among surface emissions, convective uplift, and the extent of the AMA collectively shapes the upper tropospheric methane enhancement. Meanwhile, the configuration of ASM subseasonal dynamical variability, including both the monsoon convection and AMA locations, influences the efficiency of tracer transport from the lower boundary to the upper troposphere. The WTP mode offers the most effective transport for methane-rich air from Himalayas-Gangetic Plain to the UT because the main convective sources are injected near the center location

of the AMA. The methane enhancement observed in the UT is only about 2% during the IP mode due to a horizontal displacement between its main convective source region (Himalayas-Gangetic Plain) and the AMA.

## 3.3 Contribution from the emissions and dynamics of ASM system

In Sections 3.1 and 3.2, we demonstrate that variability in UTLS methane over the Asian region is influenced by two main factors: firstly, the dynamical east-west oscillation of the ASM, which substantially modulates methane 3D-distribution and influence the efficiency of upward transport; secondly, the increase in methane emissions mainly from rice paddy cultivation in late August and early September, which potentially intensify the upper-level methane structure during the late monsoon season. In this section, we further quantify the contributions of these two factors—specifically, surface emissions and ASM dynamics—to regional methane enhancement, allowing us to assess their relative impact on methane variability in the UTLS.

Figure 5(a) presents the time series of total methane emissions (blue line) and methane concentrations in the UTLS (orange line), averaged over the ASM region (15–40°N, 15–135°E). The results indicate that the annual maxima of UTLS methane concentrations generally coincide with the peaks in annual emissions. Furthermore, years exhibiting higher emissions during the JAS compared to other years tend to correspond to elevated methane concentrations in the UTLS. For instance, during JAS 2020, when seasonal emissions reached a relatively high value (~19.3 Tg for the entire ASM region), the UTLS methane concentration also peaked (~1928 ppbv), marking the maximum throughout the simulation period.

Meanwhile, we notice that the interannual variability of methane in the UTLS (orange line) during the monsoon season is 40% larger than other seasons, while the emissions during monsoon did not show remarkably larger interannual variability than other seasons. This suggests that upper level methane during monsoon season are not solely determined by emission magnitude but to some extend influenced by the ASM dynamical conditions or other factors. For example, while the seasonal total emissions during JAS 2016 and 2018 are comparable (18.3 Tg versus 18.4 Tg), the seasonal and regional mean methane concentration in the UTLS during JAS 2018 is ~30 ppbv (~1.5%) higher than in 2016, which is a remarkable interannual difference. This difference can be attributed to variations in the distribution of anticyclone modes (Figure 5b). Specifically, during JAS 2018, the ASM anticyclone exhibited a higher frequency of WTP (western Tibetan Plateau) and ETP (eastern Tibetan Plateau) modes, particularly in August and September, creating a more favorable configuration for upward transport compared to 2016. A detailed comparison of subseasonal oscillations of the AMA between 2016 and 2018 is provided in the supplementary material (Figure S3). Note that the $CH_4$ interannual variability in the UTLS can be related to other large-scale climate modes. For example, the relative low $CH_4$ in the UTLS in 2015 is potentially attributed to suppressed updrafts influenced by ENSO-related dynamics (Alladi et al., 2024).

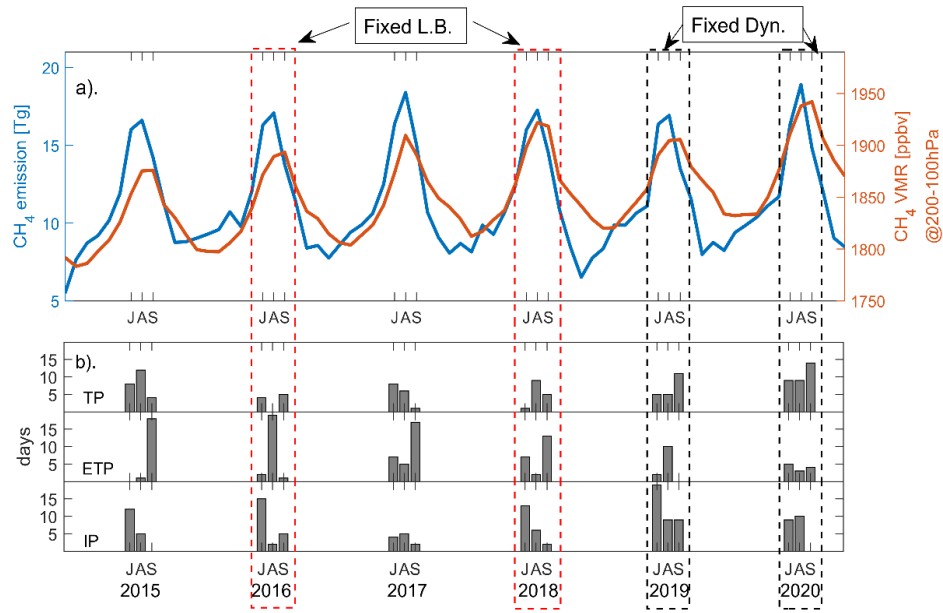

**Figure 5: (a) Time series of monthly total methane emission (unit: Tg) and VMRs (unit: ppbv) averaged over the ASM region (15-40°N, 15-135°E). (b) Number of days identified for three AMA modes during JJS for each year. The dashed red and black boxes indicate the Fixed L.B. and Fixed Dyn experiments, respectively, as described in Table 1.**

Based on the analysis above, we assess the relative contribution of emissions in the lower boundary and AMA dynamics to the upper level methane through a sensitivity test. We use the AMA dynamics of 2016 and 2018 as representative cases to examine configurations that suppress or enhance UTLS transport, respectively. To isolate the impact of AMA configuration, we conducted a test simulation using the lower boundary emissions of 2016 combined with the meteorological data of 2018 to drive the model (16LB/18Dyn). The resulting change in methane concentrations, compared to the control run for 2016

(16Ctl), represents the effect of AMA dynamics. To confirm that these results are independent of the boundary conditions, we conducted a parallel test using the lower boundary emissions of 2018 combined with the meteorological data of 2016 (18LB/16Dyn). The methane changes were then compared to the control run for 2018 (18Ctl). We found that the results of 16LB/18Dyn-16Ctl and 18Ctl-18LB/16Dyn were nearly identical (see Table 1 and Figure S4). Consequently, we use the 16LB/18Dyn - 16Ctl simulation to illustrate the influence of AMA dynamics in the following analysis.

Additionally, we use the lower boundary conditions of 2019 and 2020 as representative cases for low and high emissions over the ASM region, respectively. The total emissions over the ASM region during JAS 2020 are remarkably higher (1.3 Tg, ~7%) than those in 2019. To isolate the effect of emissions, we conducted a test simulation using the lower boundary emissions of 2020 with the meteorological data of 2019 (20LB/19Dyn) and compared the results to the control run for 2019. Similarly, we ran a test replacing the 2020 lower boundary with the 2019 emissions (19LB/20Dyn). We found that the

differences due to variations in the lower boundary emissions remained relatively consistent when using different dynamical

fields (see details in Figure S5). Therefore, in the following analysis, we use the results from 20LB/19Dyn-19Ctl to illustrate the effect of surface emissions.

**Table 1. Seasonal (JAS) and ASM regional (15-40N, 15-135E) mean CH₄ VMR (unit: ppbv) in the UTLS (200-100hPa) for the control and test runs.**

|  | Control run | | Test run | | Diff. (test-control) |
| --- | --- | --- | --- | --- | --- |
|  | Configuration | CH₄ (ppb) | Configuration | CH₄ (ppb) | ΔCH₄ (ppb/%) |
| Fixed L.B. | 16Ctl | 1883 | 16LB/18Dyn | 1903 | 20 ( 1.06%) |
|  | 18Ctl | 1909 | 18LB/16Dyn | 1888 | -21 (-1.11%) |
| Fixed Dyn. | 19Ctl | 1899 | 20LB/19Dyn | 1912 | 13 ( 0.68%) |
|  | 20Ctl | 1928 | 19LB/20Dyn | 1912 | -16 (-0.83%) |

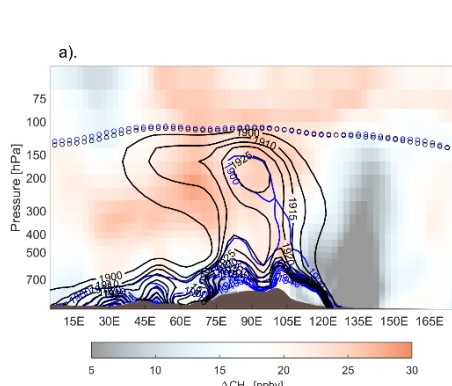

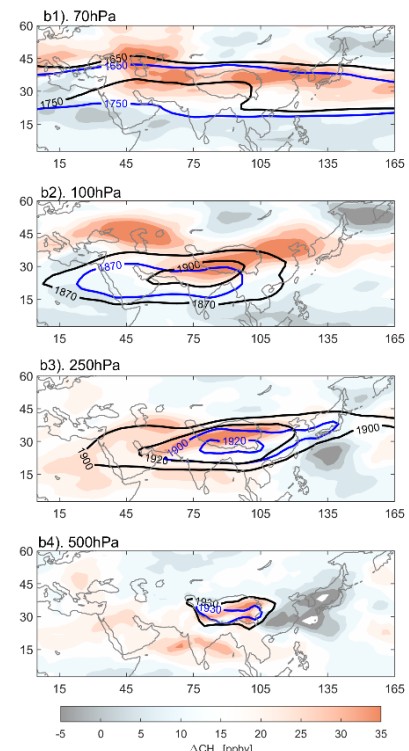

**Figure 6: The seasonal (JAS) mean differences of CH₄ VMR due to anticyclone dynamics (represented by 2016LB/2018DYN minus 2016CTL). (a) Longitude-pressure cross section of CH₄ differences (averaged over 15N-40N). The horizonal distribution of CH₄ differences are shown for 70hPa (b1), 100hPa (b2), 250hPa (b3), 500hPa (b4). Contours show the methane VMR for two runs: black for 2016LB/2018DYN and blue for 2016CTL. The circles mark the tropopause for 2016 (blue) and 2018 (black).**

Figures 6 and 7 illustrate the spatial distribution of the effects from monsoon dynamics and emission conditions, respectively. Quantitatively, monsoon dynamics introduce stronger variations in methane concentrations (1–2%) in the UTLS region

compared to the changes driven by emissions, which are less than 1%. This difference arises because the impact of dynamics is amplified in the UTLS region (Figure 6a), while the influence of emissions diminishes progressively from the surface to higher altitudes (Figure 7a).

As shown in Figure 6, changes in $CH_4$ concentrations driven by dynamical fields are significant in the mid-troposphere to lower stratosphere, with variations ranging from 10 to 40 ppbv. Notably, these dynamical-induced changes propagate toward the northern edge of the AMA in the UTLS. The dynamical-related methane anomalies extend northward into the lowermost stratosphere and eastward across the Pacific Ocean along the prevailing westerlies (see Figure S6).

In contrast, differences driven by emission conditions are more localized, concentrating in northern India and the Sichuan Basin within the lower troposphere (Figure 7b4), regions identified as critical for upward transport in Sec. 3.2. As the methane is transported upward into the upper troposphere, emission-related anomalies are redistributed and confined within the AMA region, as outlined by the GPH isolines (black contours) and do not significantly extend into the stratosphere.

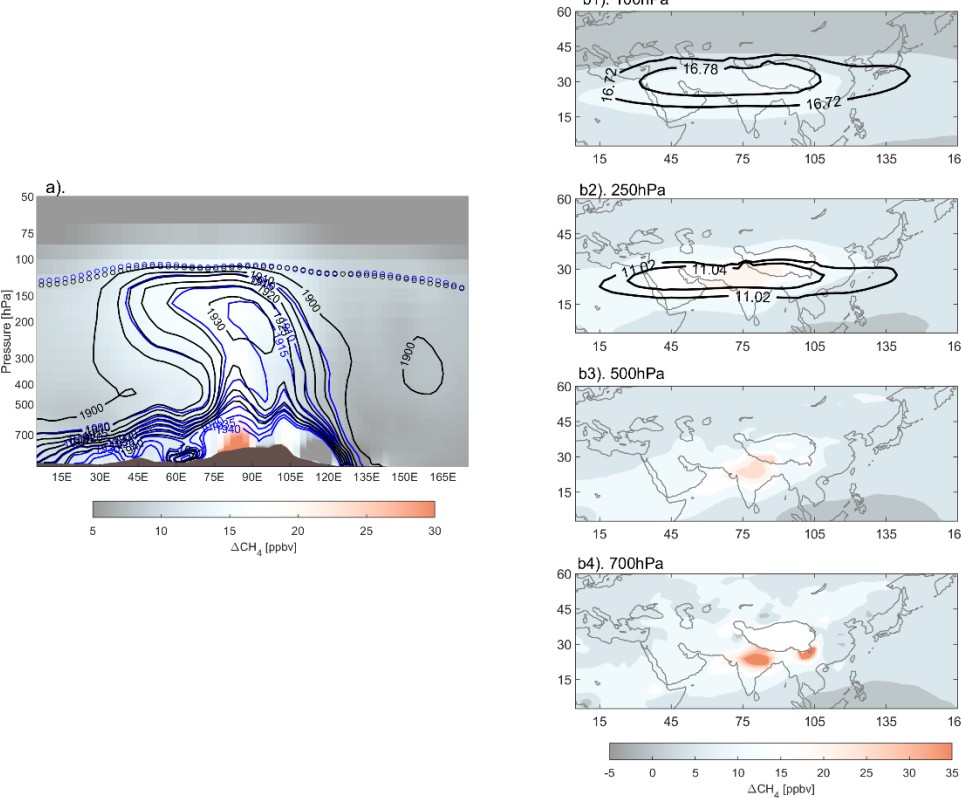

**Figure 7: Similar as Figure 6 but for differences of CH₄ VMR due to emissions (represented by 2020LB/2019DYN minus 2019CTL). Contours show the methane VMR for two runs: black for 2020LB/2019DYN and blue for 2019CTL. And the black contours on (b1) &(b2) indicate the AMA (GPH contours).**

## 4 Discussions

The model results have a reasonable representation on the $CH_4$ vertical and horizontal structure in comparison to the in-situ measurements and satellite datasets, which has been discussed in a previous study (Tao et al., 2024). The uncertainty in our simulation includes $CH_4$ emission inversion uncertainty and modelling uncertainties such as representative error (Stanevich et al., 2020, 2021), transport scheme uncertainties (Bisht et al., 2021; Saito et al., 2013) and chemistry scheme uncertainties (Murray et al., 2021; Zhao et al., 2019). Here, the *a posteriori* $CH_4$ fluxes used in our study show much larger interannual

flux variations than the corresponding *a priori* estimates, with their uncertainties (<3%) typically 40% smaller than the *a priori* uncertainties.

Our study demonstrates two key findings: (1) the subseasonal oscillations of the AMA significantly influence the methane distribution and its transport efficiency from the lower boundary to the UTLS; and (2) the interannual variations in methane enhancement in the UTLS are more strongly controlled by the conditions of AMA subseasonal oscillations than by

330 emissions alone.

Notably, regional emissions, the organized monsoon convection as well as the AMA modes are not entirely independent. On one hand, monsoon-driven heavy rainfall and resulting floods can increase methane emissions. On the other hand, the pattern of organized monsoon convection is coupled with AMA variations. One potential relationship is that upper level divergence caused by deep convection potentially shapes the AMA pattern. Therefore, further investigation is needed to understand the

335 complex interactions among monsoon convection, regional emissions, and large-scale anticyclonic circulation, as well as how these interactions evolve under climate change.

## 5 Conclusions

Similar as tropospheric tracer like CO, spatial distribution of methane in the upper troposphere exhibits remarkable subseasonal variation in strong relation to the east-west oscillation of Asian Monsoon Anticyclone (AMA). Based on AMA

mode composites, we confirm that the dynamic nature of the AMA, in terms of its subseasonal modes, modulates the horizonal distribution of methane as well as efficiency of vertical transport from the convective outflow upward to the upper troposphere over the monsoon region. In particular, the local coincidence of $CH_4$ emissions, strong convection and the location of the anticyclone around 80°E (WTP mode) favors the vertical transport of air from north India and Bangladesh to the upper troposphere, which contributes most significantly to the total $CH_4$ monsoon plume at the UT.. The methane source

from south China also contributes to the enhancement especially when AMA centering around 105°E (ETP mode). When the AMA center is located over the Iranian Plateau (around 60°E, IP mode), it is positioned far from the primary monsoon convective regions (e.g., the Indian subcontinent and the Bay of Bengal). In such a configuration, horizontal redistribution within the anticyclone becomes more dominant, rather than further vertical upwelling for these organized monsoon convective sources into the UTLS. Quantitively, the $CH_4$ anomaly in the UT under WTP mode is 50%-100% higher than that

under the other modes, which show enhanced connection from key source region Himalayas-Gangetic Plain to the upper-level AMA.

Our study further reveals that methane enhancement at the upper troposphere over Asian summer monsoon region is a joint effect of monsoon transport system and annual emission peak at late August mainly from rice cultivation. The monsoon dynamics consistently elevate upper tropospheric $CH_4$ 2%~10% throughout its whole course while emissions from rice

cultivation notably contribute to the $CH_4$ peak commonly around late August. Our model sensitivity study reveals that, as to the monsoon seasonal and regional averaged methane amplitude in the UTLS region, the influence from conditions of AMA subseasonal oscillations is more remarkable than that from conditions of emissions.

Our findings underscore the importance of monsoon dynamics and its subseasonal variability in shaping the upper tropospheric methane 3-D distribution over the Asian monsoon region. Further research is encouraged to unravel the

complexities of methane transport within the monsoon system, identify the primary source regions for methane emissions, and trace the trajectory of the monsoon methane plume after the monsoon's withdrawal.

**Data availability**

All raw data can be provided by the corresponding authors upon request.

**Code availability**

The GEOS-Chem model of atmospheric chemistry and transport model is maintained by Harvard University (http://geos-chem.seas.harvard.edu ). The ensemble Kalman filter code is publicly available at
https://github.com/Rainbow1994/EnKF_CH4.git.

**Author contribution**

MC designed the study; SZ performed the emission inversion, forward simulation and sensitivity tests; MC analyzed the data;

MT and SZ wrote the manuscript draft; All the authors reviewed and edited the manuscript.

**Competing interests**

The authors declare that they have no conflict of interest.

**Acknowledgments**

We thank the GEOS-Chem community, in particular the Harvard University team which helps maintain the GEOS-Chem
model, and the NASA Global Modelling and Assimilation Office (GMAO) for providing the MERRA2 data product.

**Financial support**

This work was supported by the National Key R&D Program of China (no. 2022YFB3904802), by the National Natural
Science Foundation of China (42105060), by the China Postdoctoral Science Foundation (E3442418), and by the Tibet
Science and Technological Project (CGZH2023000385).

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
