# Peer review of "Significant Response of Methane in the Upper Troposphere to Subseasonal Variability of the Asian Monsoon Anticyclone"

_EGUsphere, 2024_

## Author Response (AR1)

**Response to Referee #1**

We thank Referee 1 for the thoughtful and detailed comments and suggestions, which definitely helped us to improve the manuscript. We addressed all points in the revised version. Reviewer comments are in black, answers in blue. Page and line numbers from the updated manuscript are underlined. The main changes in the revised version are:

- We have clarified the role of the monsoon dynamical system in methane transport, with particular emphasis on the rapid vertical lifting by deep monsoon convection and the influence of the relative position between the AMA and organized monsoon convection during the slow upwelling from the outflow level to the UTLS region. To better illustrate this, we have expanded the relevant background discussion on rapid lifting and slow upwelling in the Introduction, thoroughly revised Section 3.2, and added panels on deep convection and lower tropospheric methane to Figure 4.

- To improve the description of the AMA mode classification, we have extensively revised Section 2.2 and added Figure 1 to more intuitively illustrate the spatial morphology of different modes and the statistical distribution of anticyclone centers.

- Several previously ambiguous terms in the manuscript, such as the definitions of "anomalies" and "enhancement," as well as the earlier inadequate use of "transport pathway," have been revised for greater accuracy and clarity.

The authors simulated upper tropospheric methane distributions using GEOS-Chem driven with "optimized" surface methane flux data and showed strong subseasonal variation linked to the east-west oscillation of the Asian monsoon anticyclone (AMA). What is new about this study is twofold, the use of a surface methane flux dataset optimized with observational data and the finding that upper tropospheric methane concentrations peaked in September due to rice paddy emissions from Southeast Asia. Extensive work has been conducted to examine the AMA's role in redistributing CO, water vapor, HCN, hydrocarbons and aerosols in the upper troposphere, as summarized in the Introduction, and this paper added $CH_4$ into that list. The study clearly shows how subseasonal variations in upper tropospheric $CH_4$ over the Iranian Plateau to the

Tibetan Plateau align with AMA variability and surface emissions. Below are my comments.

First, my technical review suggested that the authors provide a clear definition of "anomaly" at the outset, as the term requires a reference point, particularly in climatology. While this may be self-evident to the authors, an explicit definition would benefit ACP's broader readership. However, such a definition is still missing. Moreover, their "anomaly" appears synonymous with "enhancement." For consistency, the authors should consider using one term throughout. In cases where they quantify upper tropospheric "enhancement" over the Asian monsoon region (lines 64–65), they must clarify the reference used to determine these values, preventing ambiguity for readers.

Thank you very much for the insightful suggestion. We have added a detailed description of the definitions of "anomaly" and "enhancement" when we use this term and ensure reproducibility. For example, the "enhancement" used in P3 L74-75 is revised as "…This increase of $CH_4$ is about 100 ppb higher than its regional annual mean volume mixing ratio (VMR) and is 3%~10% higher than the VMRs averaged over the same latitude (Tao et al., 2024; Xiong et al., 2009). " In addition, we have revised **the captions for Figure 2 and Figure 4** to include brief explanations of these terms, providing readers with clearer context when interpreting the figures.

Second, the case study seems unnecessary. The authors should move directly to the composites of the AMA modes and their corresponding methane distributions. It is also unclear whether they identified AMA modes independently and what methodology they used.

Thank you for your kind reminder. In the revised manuscript, we have added a new figure (Figure 1) along with a corresponding description in the Methods section to clarify how the AMA modes were identified and to detail the methodology employed. Additionally, we retained the case analysis—now presented as a combined figure (original Figures 1 and 2)—at the beginning of the Results section to provide readers with a clear and informative introduction to the three AMA modes, based on actual events observed during the study period. We believe this part is necessary to illustrate

the covariability of the AMA and the horizonal distribution of methane in the UTLS on daily basis.

**Specific comments:**

1.  Lines 116–127: When describing the probability distribution of AMA center positions and classifying AMA modes, including one or two figures would improve clarity. Additionally, the statement in lines 121–122 requires a concise summary of the different effects of AMA modes. This would not only make the paper more self-contained but also help explain the differing distributions of upper tropospheric methane concentrations associated with each AMA mode.

    Thanks a lot for your suggestions. We have added a figure illustrating the horizontal distribution of wind and geopotential height for the three AMA modes, along with a detailed explanation in the revised Method*s* section.

2.  Study period: The study period should be stated upfront rather than buried near the end of Section 2.

    Thanks for your recommendation. We've emphasized the study period in the first sentence of Section 2 (P3 L87) and mentioned this period again at P5 L129.

3.  Lines 132: Figure 1 – Explain what $\Delta CH_4$ in Figure 1b represents.

    Thank you for the kind reminder. We have added explanation of $\Delta CH_4$ in revised text (P7 L171-172) and revised caption of new Figure 2b (P8 L186-193).

    "Hovemöller diagrams of anomalies in (a) geopotential height and (b) CH4 concentrations at 150hPa for JAS 2020. Anomalies are calculated with respect to the daily mean values averaged over the main ASM region (15°N–40°N, 15°E–135°E)."

4.  Lines 133-134: The statement about thermal heating in the Tibetan Plateau (TP) and subsequent westward migration due to instability needs supporting references or evidence.

We rewrote this paragraph and remove this information about "thermal heating" because it is not relevant to our main storyline "the covariability of the large-scale circulation of ASM and the horizontal distribution of CH$_4$ at 150 hPa on daily basis". Please check the first paragraph in section 3.1 (P7 L169-185).

5. Line 134: "the AMA center over predominantly hovered east of 75°E" – It doesn't read right.

We have revised this sentence with "the AMA center remained predominantly east of 75°E." (P7 L179-180).

6. Line 135: Provide a brief explanation of how the AMA position influences the ΔCH$_4$ distribution.

We revised this sentence to: "This subseasonal variability of AMA significantly modulated the CH$_4$ variations in the middle to upper troposphere (see Figure 1b), similar to the behavior of tracers such as CO (Pan et al., 2016), primarily due to anticyclonic confinement (Ploeger et al., 2015)."

7. Lines 146-148: Clarify what is meant by "stirring interaction." Additionally, explain the mechanism by which boundary layer air is lifted, as this is key to transporting surface methane emissions into the upper troposphere. The authors should support these statements with references or their own evidence.

We agree that this point was not supported sufficiently. We added a subfigure in Fig. S1 to show the CH$_4$ VMR near surface with OLR pattern representing the deep convection. Here we revised this sentence to: "For example, at IP mode on Aug. 12st, the high methane center locates southern edge of AMA. This pattern results from the "stirring" interaction between convection-uplifted boundary layer air from the Indian subcontinent (as shown by the overlap between the main monsoon convection source and high methane regions near the lower boundary in bottom panel of Fig. S1) and the surrounding air, which is similar like "stirring" interaction proposed by Pan et al. (2016)." (P9 L205-208).

8. Line 176: The term "pathway" should be reconsidered unless streamlines or variables indicating a dynamic process are presented.

We recognize that our use of "transport pathway" was somewhat misleading. In response, we have revised the text to: "the subseasonal oscillations of the AMA significantly influence the methane distribution and its transport efficiency from the lower boundary to the UTLS." (P20_L198). Additionally, we have replaced the inappropriate use of "transport pathway" throughout the manuscript.

9. Line 198: The phrase "(unit: %, referring to zonal mean)" is unclear. The authors should explicitly define how anomalies were calculated (per the first comment) and clarify what is being presented. It appears they are referring to zonal means of anomalies within their domain.

The definition of "anomalies" is unclear. Here the relative anomalies refer to global zonal mean at each latitude within each pressure layer for corresponding mode composite. To be clearer, we revised the figure caption (P15 L280-287). In addition, we added one figure with unit in absolute values of VMR (Fig. S3) as a supplement.

10. The anomalies are more precisely relative anomalies referring to the zonal mean methane VMR within the domain (…). Add Figure in ppbv as a supplement.

Thanks for your suggestion. We added a figure in ppbv has been added in the revised supplementary file as Figure S3.

**Response to Referee #2**

This is an interesting and important study, which merits its publication in ACP. The

scientific content, the quality of the study and its presentation is good, however I suggest

some revisions before publication by ACP.

General comment:

My principle concern is that the role of convection within the study by Zhu et al. (2025) is not discussed sufficiently. 'The complex interaction between monsoon dynamics and surface emissions to determine the upper tropospheric methane' is highlighted as main result of the paper. However, the definition of 'monsoon dynamics' remains unclear. My impression is that 'monsoon dynamics' stands here for the spatial-temporal variability of the Asian summer monsoon anticyclone in the upper troposphere and lower stratosphere (east-west and south-north shift, Iranian and Tibetan mode). However, convection that uplift methane to altitudes of the upper troposphere plays a major role within the monsoon dynamics. Maybe there is a misunderstanding, therefore, I recommend improving the study by clarify the role of convection. More specific comments to this issue will follow below.

We sincerely thank Referee #2 for the thoughtful and constructive comments, which have significantly improved the quality and clarity of our manuscript. We have carefully addressed all points raised in the revised version. Reviewer comments are presented in black, with our responses provided in blue. Page and line numbers from the updated manuscript are underlined. The main revisions made to the manuscript are summarized below:

- We have clarified the role of the monsoon dynamical system in methane transport, with particular emphasis on the rapid vertical lifting by deep monsoon convection and the influence of the relative position between the AMA and organized monsoon convection during the slow upwelling from the outflow level to the UTLS region. To better illustrate this, we have expanded the relevant background discussion on rapid lifting and slow upwelling in the Introduction, thoroughly revised Section 3.2, and added panels on deep convection and lower tropospheric methane to Figure 4.
- To improve the description of the AMA mode classification, we have extensively revised Section 2.2 and added Figure 1 to more intuitively illustrate the spatial morphology of different modes and the statistical distribution of anticyclone centers.
- Several previously ambiguous terms in the manuscript, such as the definitions of "anomalies" and "enhancement," as well as the earlier inadequate use of "transport pathway," have been revised for greater accuracy and clarity.

**Major comments:**

1. p1 L26: 'The AMA center over the Iranian Plateau suppresses the vertical transport'. This formulation is confusing. Over the Iranian Plateau shallower and less intense convection occurs during summer compared to regions further east (e.g Indian subcontinent, Bay of Bengal, China). Therefore, a lower amount (or rather no) methane can be transported from surface levels to the upper troposphere over the Iranian Plateau (see Fig. 3 in Zhu et al., 2025). Thus, if the anticyclone is over IP far off the strong convective sources (e.g. Indian subcontinent, Bay of Bengal, China), methane cannot be uplifted locally over the Iranian Plateau into altitudes of the anticyclone in contrast to the TP mode. What is meant with 'suppresses the vertical transport'? That the location of the AMA over the Iranian Plateau suppress convection over the Iranian Plateau? Please clarify.

   We thank the reviewer for pointing out the ambiguity in our original formulation. We agree that the previous statement "The AMA center over the Iranian Plateau suppresses the vertical transport" was not sufficiently precise and may have led to misunderstanding. Our intended meaning was that when the AMA center is located over the Iranian Plateau, it is positioned far from the primary monsoon convective regions (e.g., the Indian subcontinent and the Bay of Bengal), and therefore it suppresses the vertical transport of air from these organized monsoon convective sources—specifically from their convective outflow levels to higher altitudes in the upper troposphere and lower stratosphere (UTLS). In such a configuration, horizontal redistribution within the anticyclone becomes more dominant, rather than further vertical uplift into the UTLS.

   We have revised the relevant sentence in the abstract to clarify this point as follows:

   "The AMA center around 80°E favors the upward transport from organized monsoon convective sources over the Indian subcontinent and Bay of Bengal while the AMA center around 105°E favors the source from southwest China transported to the upper troposphere. When the AMA shifts over the Iranian Plateau, vertical transport from the convective outflow level further to the upper troposphere is weakened and the horizontal redistribution becomes dominant."

And we also rewrote the relevant parts in sec. 3.2 first paragraph in the conclusion in a much clearer way, which is also responsible for 15th and 16th major comments below.

2.  p2 L40: '(Park et al., 2009; Pan et al., 2016; Randel et al., 2010; Rosenlof et al., 1997; Yu et al., 2017)' Please add here some more recent publications e.g. from aircraft campaigns that measured air within the AMA during StratoClim 2017 and ACCLIP 2022 showing enhanced tropospheric tracers within the anticyclone.

We thank the reviewer's suggestion. These recent flight campaigns nicely supported this point. The following relevant references have been added in P2 L43 of the revised manuscript:

[1] Pan, L. L., Atlas, E. L., Honomichl, S. B., Smith, W. P., Kinnison, D. E., Solomon, S., Santee, M. L., Saiz-Lopez, A., Laube, J. C., Wang, B., Ueyama, R., Bresch, J. F., Hornbrook, R. S., Apel, E. C., Hills, A. J., Treadaway, V., Smith, K., Schauffler, S., Donnelly, S., … Newman, P. A. (2024). East Asian summer monsoon delivers large abundances of very short-lived organic chlorine substances to the lower stratosphere. Proceedings of the National Academy of Sciences, 121(12). https://doi.org/10.1073/pnas.2318716121

[2] Bucci, S., Legras, B., Sellitto, P., D'Amato, F., Viciani, S., Montori, A., Chiarugi, A., Ravegnani, F., Ulanovsky, A., Cairo, F., & Stroh, F. (2020). Deep-convective influence on the upper troposphere–lower stratosphere composition in the Asian monsoon anticyclone region: 2017 StratoClim campaign results. Atmospheric Chemistry and Physics, 20(20), 12193–12210. https://doi.org/10.5194/acp-20-12193-2020

3. p2 L2: 'with a vertical velocity of about 1–1.5 K/day' This value depends on the used reanalysis. 1–1.5 K/day is related to ERA-Interim. Please clarify.

We appreciate the reviewer's comment and agree that the vertical velocity value of 1–1.5 K/day is specific to ERA-Interim (Vogel et al.,2019; Ploeger et al., 2017). The original sentence did not make this clear. Indeed, this estimate is based on previous studies using ERA-Interim (e.g., Schoeberl et al., 2012; Garny and Randel, 2016), which have shown to overestimate the diabatic heating rate by about 30–40%, likely due to longwave radiative heating biases and a cold temperature bias in the tropical tropopause layer (TTL) (Wright and Fueglistaler, 2013). This implies that the more realistic upwelling rates should be in the range of ~0.3–0.8 K/day, consistent with measurement-based evaluations (e.g., von Hobe et al., 2021). Moreover, its newer generation ERA5 provide a more accurate representation of ASM transport processes. ERA5 shows stronger convective detrainment, especially over the Tibetan Plateau, and slower large-scale upwelling above 380 K compared to ERA-Interim (Legras and Bucci, 2020; Vogel et al., 2024). The overall upwelling in ERA5 is generally slower, aligning better with in-situ and satellite-based estimates.

In response, we have revised this estimate of upwelling to '~0.3–0.8 K/day' and added following references. Additionally, we have revised this paragraph in the introduction to better descript the monsoon transport system, which includes fast convective transport from the boundary layer to the outflow level, and the slower, large-scale ascent from the outflow level to the UTLS (P2 L55). We hope this improves the clarity of the ASM vertical transport structure.

[1] Wright, J. S. and Fueglistaler, S.: Large differences in reanalyses of diabatic heating in the tropical upper troposphere and lower stratosphere, Atmos. Chem. Phys., 13, 9565–9576, https://doi.org/10.5194/acp-13-9565-2013, 2013.

[2] Garny, H. and Randel, W. J.: Transport pathways from the Asian monsoon anticyclone to the stratosphere, Atmos. Chem. Phys., 16, 2703–2718, https://doi.org/10.5194/acp-16-2703-2016, 2016.

[3] Legras, B. and Bucci, S.: Confinement of air in the Asian monsoon anticyclone and pathways of convective air to the stratosphere during the summer season, Atmos. Chem. Phys., 20, 11045–11064, https://doi.org/10.5194/acp-20-11045-2020, 2020.

[4] von Hobe, M., Ploeger, F., Konopka, P., Kloss, C., Ulanowski, A., Yushkov, V., Ravegnani, F., Volk, C. M., Pan, L. L., Honomichl, S. B., Tilmes, S., Kinnison, D. E., Garcia, R. R., and Wright, J. S.: Upward transport into and within the Asian monsoon anticyclone as inferred from StratoClim trace gas observations, Atmos. Chem. Phys., 21, 1267–1285, https://doi.org/10.5194/acp-21-1267-2021, 2021.

[5] Vogel, B., Volk, C. M., Wintel, J., Lauther, V., Clemens, J., Grooß, J.-U., Günther, G., Hoffmann, L., Laube, J. C., Müller, R., Ploeger, F., and Stroh, F.: Evaluation of vertical transport in ERA5 and ERA-Interim reanalysis using high-altitude aircraft measurements in the Asian summer monsoon 2017, Atmos. Chem. Phys., 24, 317–343, https://doi.org/10.5194/acp-24-317-2024, 2024.

4. p3 L3: 'The debate persists over whether the seasonal increase of UT methane in the Asian monsoon region is due to enhanced summer emissions from regional rice paddies (Zhang et al., 2020) or the upward transport by the monsoonal circulation'. Could it also be the combination of both?

Yes, we agree that both enhanced regional emissions and monsoonal transport contribute to the observed seasonal increase in upper tropospheric methane. In fact, our conclusion also supports the role of both mechanisms (dynamical effect dominant). The original sentence in the Introduction may not have accurately reflected this nuance. Our intended point was to highlight that the relative importance of these two processes remains under debate. To clarify this, we have revised the sentence as follows:

"The debate persists over which factor plays the dominant role in the seasonal increase of UT methane over the Asian monsoon region—enhanced summer emissions from regional rice paddies (Zhang et al., 2020) or the strong upward transport by the monsoon convection and circulation (Zeng et al., 2021) —as both are known to contribute."

5. p3 Sect. 2.1: Please provide some information about the treatment of convection in the GEOS-Chem model. Is there an additional convection scheme included or is just used the convection included in MERRA-2?

The treatment of convection in GEOS-Chem is not simulated directly within the model but is instead derived from the meteorological fields provided by the MERRA-2 reanalysis. This explanation has been added in the revised manuscript (P4 L97).

6. p4 Sect. 2.2: A map showing the regions of the different modes (IP, WTP, ETP) would be very helpful.

Thanks for your suggestion. In the revised manuscript, we have added a new figure (Figure 1) along with a corresponding description in the Methods section to clarify how the AMA modes were identified and to detail the methodology employed (P6 L150-157).

7. p6 Sect. 3.1: Please provide a map showing the regional/spatial distribution (fluxes) of $CH_4$ emission in Asia in the model at surface.

Note that the distribution $CH_4$ fluxes has been provided in the supplementary Figure S1. In addition, we added a subfigure to show the map of near-surface methane VMR and low OLR contours (representing deep convection).

8. p5 L131: 'shows the anomalies of GPH as well as methane on 150 hPa' Please explain how 'anomalies' are defined / calculated here.

Thank you for the kind reminder. We have revised the caption of the new Figure 2 to include an explanation of the "anomalies of GPH and methane at 150 hPa." and descried it the revised text (P7 L170-171).

p6 L157: 'The timing of this CH4 surge aligns closely with the seasonal emissions peak from rice paddy cultivation.' Please provide more information about subseasonal variability of CH4 at model boundary layer

Thank you very much for the suggestion. This information is mainly based on Fig. 3 in Zhang et al. (2020). They used Modis-based Enhanced vegetation index (EVI) to

estimate the growth of rice paddies. And we saw its seasonal peaks mainly at late August. Thus, we added this reference here. In fact, our Figure 5 also show the emission peaks at middle to late August (mainly from agriculture section but not shown).

9. p7 L176: 'Under IP mode, the horizontal redistribution is remarkable with a weak vertical pathway.'

I think this statement should be made a bit clearer such as: 'Over the Iranian Plateau shallower and less intense convection occurs during summer compared to regions in WTP and ETP, therefore CH4 enhancements over the IP are caused by horizontal westward transport in the UT caused by the AMA and not by local vertical transport.'

Thank you for your suggestion. We have followed your advice and now present the OLR pattern for each mode in the new Figure 4. However, the statement that "shallower and less intense convection occurs over the Iranian Plateau during summer compared to regions in the WTP and ETP" does not totally true in our case. Our results show that, during the IP mode, the deep convection pattern shifts slightly westward and is also intensified over the South China Sea. The overall strength of convection, as interpreted from OLR, during the IP mode is comparable to that in the WTP and ETP modes.

Therefore, when considering both the convection patterns and the 3D methane distribution, our findings indicate that, when the AMA center is located over the Iranian Plateau, it is positioned farther from the primary monsoon convective regions (e.g., the Indian subcontinent and the Bay of Bengal). As a result, vertical transport of air from these organized monsoon convective sources into the upper troposphere and lower stratosphere (UTLS)—specifically from their convective outflow levels to higher altitudes—is suppressed. In this configuration, horizontal redistribution within the anticyclone becomes more dominant than further vertical uplift into the UTLS.

This also addresses the first major comment.

10. p9 L207: 'we demonstrate that variability in UTLS methane over the Asian region is influenced by two main factors: firstly, the dynamical east-west oscillation of the ASM, which substantially modulates methane distribution and influence the upward transport

pathways; secondly, the increase in methane emissions from rice paddy cultivation in late August and early September'

I am missing in this discussion the role of convection. The upward transport depends on the location of strong convection. If the AMA anticyclone is over convective regions enhanced CH4 can be injected into the AMA (WTP, ETP), subsequently by horizontal displacement enhanced CH4 can be transported horizontally westward (IP) here no local injection of CH4 into the AMA by convection occurs.

Thanks for the suggestion. We deeply revised sec. 3.2. Firstly, a row of subfigures is added to Fig. 4 to show the pattern of deep convection (OLR), near-surface winds as well as methane in the lower troposphere. And accordingly, we rewrote this section to discuss the role of convection and the interplay between the convection and large-scale circulation (P10-16 L229-303).

11. p10 L211:  'ASM dynamics' = convection + horizontal shift of AMA?

Yes, you are correct. In general, three main factors determine UTLS methane levels: surface emissions, convective transport, and the large-scale circulation associated with the ASM. We use the term "ASM dynamics" to collectively refer to both convective transport and large-scale circulation. In revised version, we explicitly describe "ASM dynamics" before this at P16 L295-298: 'the configuration of ASM subseasonal dynamical variability, including both the monsoon convection and AMA locations, influences the efficiency of tracer transport from the lower boundary to the upper troposphere'. Additionally, the variability of monsoon convection and the AMA are interrelated, this point is further addressed in the discussion section (P21 L417-421).

12. p10 L220-225: 'What about differences in strength and location of convection in your model between the different years from 2015 to 2020 such as modulations by ENSO?

Alladi et al (2024) shows, that during Asian monsoon 2015, a substantial reduction of the tropospheric species at 100 hPa and 146 hPa in contrast to 2022 where observed mainly attributed to the weaker updrafts in 2015 owing to strong El Nino conditions.

Hemanth Kumar Alladi, P.R. Satheesh Chandran, Venkat Ratnam M, Impact of ENSO on the UTLS chemical composition in the Asian Summer Monsoon Anticyclone, Atmospheric Research, Volume 309, 2024, 107551, ISSN 0169-8095, https://doi.org/10.1016/j.atmosres.2024.107551.

(https://www.sciencedirect.com/science/article/pii/S0169809524003338)

The lower CH4 mixing rations in 2015 shown in Fig. 5 (Zhu et al., 2025) could be caused by weaker updrafts in 2015.

We thank the reviewer for this important point. As noted by Alladi et al. (2024), the 2015 Asian Summer Monsoon season, which coincided with a strong El Niño event, exhibited noticeably weaker convective updrafts compared to neutral or La Niña years such as 2022. This reduction in convection led to substantially lower concentrations of tropospheric trace species—including methane—at 100 hPa and 146 hPa, consistent with our model results shown in Fig. 5. Thus, it worth to mention this point: "Note that the CH$_4$ interannual variability in the UTLS can be related to other large-scale climate modes. For example, the relative low CH$_4$ in the UTLS at 2015 is potentially attributed to suppressed updrafts influenced by ENSO-related dynamics (Alladi et al., 2024)." (P17 L331-333).

13. p13 L284-L287: 'the subseasonal oscillations of the Asian Monsoon Anticyclone (AMA) significantly influence the methane transport pathway and its efficiency from the lower boundary to the UTLS (upper troposphere and lower stratosphere' This sentence is somewhat misleading. Please clarify 'transport pathways'.

We appreciate your feedback. We recognize that our use of "transport pathway" was somewhat misleading. In response, we have revised the text to: "the subseasonal oscillations of the AMA significantly influence the methane distribution and its transport efficiency from the lower boundary to the UTLS."(P20 L398). Additionally, we have replaced the inappropriate use of "transport pathway" throughout the manuscript.

14. p14 L295: 'Based on AMA mode composites, we confirm that the dynamic nature of the AMA, in terms of its subseasonal modes, modulates the vertical structure of CH4 over the monsoon region.' --> modulates the horizontal structure.   Please clarify.

This sentence, similar as few places with a similar statement, is revised as: "we confirm that the dynamic nature of the AMA, in terms of its subseasonal modes, modulates the horizonal distribution of methane as well as efficiency of vertical transport from the convective outflow upward to the upper troposphere over the monsoon region." (P21 L410-412).

15. p14 L296: 'In particular, AMA centering around 80°E (WTP mode) favors the vertical transport of air from north India and Bangladesh to the upper troposphere, which contributes most significantly to the total CH4 monsoon plume at the UT.' -->

'The local coincidence of CH4 emissions, strong convection and the location of the anticyclone around 80°E (WTP mode) favors the vertical transport of air from north India and Bangladesh to the upper troposphere, which contributes most significantly to the total CH4 monsoon plume at the UT.'

We appreciate this nice suggestion and take this sentence in our revised manuscript (P21 L412-415).

**Minor/technical comments:**

1.  p2 L34: 'upper atmosphere' -> 'upper troposphere' or 'UTLS'

    Revised.

2.  p4 L114: 'tracer transport' -> 'transport of trace gases'

    Revised.

3.  p4 L117: 'Our statistics show that the frequency of the Tibetan mode (the eastern phase of the distribution) is nearly twice that of the Iranian mode.' Please specify the time period for the statistics. 2015-2020?

Done.

4. p5 L130: 'on the vertical distribution of CH4 in 2020 summer' -> 'on the horizontal distribution of ... at 150hPa'

   Done.

5. p6 L157: 'in Figure.1' -> 'in Figure 1'

   Done.

6. p7 L176: 'horizonal' -> 'horizontal'

   Done.

7. p7 L179: 'in Figure.S1' -> 'in Figure S1'

   Done.

8. p10 L216: 'late-summer emissions' -> 'higher emission during the JAS than in other years tend to ....'

   Done.

9. Fig. 5: The red and blue lines are very thin. Please make them a bit thicker for better visibility.

   Done.

10. Add in the figure caption the meaning of the dashed red and black boxes that are related to table 1.

    Revised.

11. Tab. 1: Please add CH4 [ppbv] as subtitle in the table.

    Revised.